# Unraveling the Nrf2-ARE Signaling Pathway in the DF-1 Chicken Fibroblast Cell Line: Insights into T-2 Toxin-Induced Oxidative Stress Regulation

**DOI:** 10.3390/toxins15110627

**Published:** 2023-10-25

**Authors:** Suisui Gao, Kaixin Wang, Kuankuan Xiong, Shuai Xiao, Chujian Wu, Mingxia Zhou, Linfeng Li, Guoxiang Yuan, Lihuang Jiang, Qianbo Xiong, Lingchen Yang

**Affiliations:** College of Veterinary Medicine, Hunan Agricultural University, No. 1 Nongda Road, Furong District, Changsha 410128, China; gaoss@stu.hunau.edu.cn (S.G.); wangkaixin0708@stu.hunau.edu.cn (K.W.); kuankuanxiong@stu.hunau.edu.cn (K.X.); shuaixiao@stu.hunau.edu.cn (S.X.); wuchujian@ringpai.com (C.W.); mingxiazhou@stu.hunau.edu.cn (M.Z.); 13407319966@stu.hunau.edu.cn (L.L.); yuanguoxiang@stu.hunau.edu.cn (G.Y.); jianglihuang@stu.hunau.edu.cn (L.J.); bobx972@stu.hunau.edu.cn (Q.X.)

**Keywords:** T-2 toxin, oxidative stress, Nrf2-ARE signaling pathway, antioxidant factor, DF-1 cells

## Abstract

The T-2 toxin (T2) poses a major threat to the health and productivity of animals. The present study aimed to investigate the regulatory mechanism of Nrf2 derived from broilers against T2-induced oxidative damage. DF-1 cells, including those with normal characteristics, as well as those overexpressing or with a knockout of specific components, were exposed to a 24 h treatment of 50 nM T2. The primary objective was to evaluate the indicators associated with oxidative stress and the expression of downstream antioxidant factors regulated by the Nrf2-ARE signaling pathway, at both the mRNA and protein levels. The findings of this study demonstrated a noteworthy relationship between the up-regulation of the Nrf2 protein and a considerable reduction in the oxidative stress levels within DF-1 cells (*p* < 0.05). Furthermore, this up-regulation was associated with a notable increase in the mRNA and protein levels of antioxidant factors downstream of the Nrf2-ARE signaling pathway (*p* < 0.05). Conversely, the down-regulation of the Nrf2 protein was linked to a marked elevation in oxidative stress levels in DF-1 cells (*p* < 0.05). Additionally, this down-regulation resulted in a significant decrease in both the mRNA and protein expression of antioxidant factors (*p* < 0.05). This experiment lays a theoretical foundation for investigating the detrimental impacts of T2 on broiler chickens. It also establishes a research framework for employing the Nrf2 protein in broiler chicken production and breeding. Moreover, it introduces novel insights for the prospective management of oxidative stress-related ailments in the livestock and poultry industry.

## 1. Introduction

The T-2 toxin (T2), a highly toxic member of the mycotoxin group [1], has been categorized as a hazardous natural pollutant by the World Health Organization (WHO), being included in the list of the most perilous pollutants in 1974 [2,3]. Varied degrees of T2 contamination have been documented in global stocks of maize, wheat, barley, and rice during storage [4,5]. Even at low contamination levels, the potential risk to both human and animal well-being remains significant. Previous studies have demonstrated that T2 is highly toxic to the digestive and immune systems of poultry [6,7]. In recent years, the application of microbial detoxification techniques has gained considerable attraction for the mitigation of mycotoxins in food and animal feed [8].

T2 exposure triggers oxidative stress and cellular harm in vivo, subsequently impairing animal health and performance [9,10,11]. The nuclear factor E2 related factor 2—antioxidant response element (Nrf2-ARE) signaling pathway plays a critical role in defending cells against oxidative stress (Figure 1), with Nrf2 constituting a pivotal constituent of this protective mechanism. It has been shown that 5–1000 nM T2 can significantly reduce the activity of chicken primary hepatocytes and chicken growth plate chondrocytes [12,13]. Sun et al. showed that T2 displayed dose and time-dependent toxic effects on mouse microglia, implicating the generation of reactive oxygen species (ROS), initiation of the mitochondrial apoptosis pathway, and repression of the Nrf2/Heme oxygenase 1 (HO-1) pathway [14]. Meanwhile, Shifrin et al. observed rapid apoptosis in Jurkat T cells upon T2 treatment, attributed to the activation of c-Jun N-terminal kinase and p38 mitogen-activated protein kinase [15]. Furthermore, Yin et al. identified T2 as an inducer of mitochondrial-mediated apoptosis in chicken hepatocytes, facilitated through ROS production, promoting cytochrome c translocation and apoptotic body formation, and triggering autophagy via the modulation of phosphatidylinositol 3-kinase (PI3K)/AKT/mammalian target of the rapamycin (mTOR) signaling pathway proteins [16,17].

Nrf2, a pivotal component in cellular antioxidation, was initially identified from the K592 cDNA library [18]. It exhibits wide distribution across diverse animal tissues and organs, including the kidney, liver, lung, digestive tract, and skin, where it displays prominent expression levels [19]. Nrf2 and Bach1 bind to the ARE, activating relevant antioxidant enzymes and protecting the normal functions of tissues and cells [20]. Under oxidative stress conditions, Nrf2 enters the nucleus and forms a heterodimer with the MAF transcription factor (Maf) protein, ultimately binding to ARE-dependent genes to exert antioxidant effects [21]. Ordinarily quiescent, Nrf2 remains sequestered in its dimeric form with kelch-like ECH-associated protein 1 (Keap1) within the cytoplasm. However, it can swiftly undergo degradation through the ubiquitin–proteasome pathway, thereby maintaining its subdued transcriptional activity [22]. The activation of the Nrf2-Keap1/ARE signal pathway enhances the body’s antioxidant stress function by promoting the transcription of downstream protective genes. This antioxidant enzyme system effectively counteracts excessive free radicals and oxidized proteins, thus averting oxidative stress and mitigating oxidative damage stemming from ROS or electrophiles.

Ingestion of T2 by animals and humans can have serious toxic effects, which in turn affect the sustainability of the entire livestock industry. To date, there is no established remedy for countering the toxic ramifications of T2. Despite considerable endeavors to comprehend the cellular toxic responses triggered by T2 exposure, the underlying mechanisms of T2-induced oxidative stress remain to be fully elucidated. The impact of Nrf2 on regulating oxidative stress induced by T2 has received limited attention. The present study aimed to address this gap in knowledge by investigating the effect of broiler-derived Nrf2 protein on T2-induced oxidative damage, through the construction of Nrf2 overexpression and knockdown cell lines. This study contributes to a more profound comprehension of the toxic effects of T2 on broilers and provides a foundation for exploring the potential use of the Nrf2 protein in broiler production and breeding, as well as new ideas for the treatment of poultry diseases related to oxidative stress.

## 2. Results

### 2.1. Toxicity Effects of T2 in DF-1 Cells

The toxicity of T2 on DF-1 cells was investigated and the results are presented in Figure 2A–F. The results showed that the number of DF-1 cells significantly decreased in a concentration-dependent manner upon exposure to T2. The cells appeared wider apart, had reduced connections, and were more crumpled, leading to an increase in unadhered and floating dead cells, as well as cell debris.

The cell viability and LDH leakage rate of DF-1 cells were also affected by T2 exposure, as indicated in Figure 2G,H. At 24 h, treatment with 50 nM and 100 nM T2 resulted in a significant increase in cell viability compared to the control cells (*p* < 0.05), whereas there were no significant differences in the 1 nM and 10 nM T2-treated groups (*p* > 0.05). However, at 48 h, the 20 nM, 50 nM, and 100 nM T2-treated groups showed a more significant decrease in cell viability compared to the control group (*p* < 0.05). No significant differences in cell viability were observed between the control and 10 nM T2 dose groups (*p* > 0.05).

Figure 2I,J depicted variations in the LDH leakage rate of DF-1 cells treated with different doses of T2 for 24 and 48 h. At 24 h, treatment with 50 nM and 100 nM T2 resulted in a significant increased the leakage rate of LDH in cells (*p* < 0.05). At 48 h, compared with the control group (0 nM), the LDH leakage rate significantly decreased at 10 nM (*p* < 0.05), while at 20, 50, and 100 nM, the LDH leakage rate significantly increased (*p* < 0.05), and at 100 nM, the cell LDH leakage rate was the highest (*p* < 0.05). After a 24 h treatment, it was observed that the application of 50 nM and 100 nM T2 resulted in a substantial increase in LDH leakage rates in the cells (*p* < 0.05). However, at the 48 h mark, in comparison to the control group (0 nM), LDH leakage rates exhibited a significant reduction at 10 nM (*p* < 0.05). Conversely, at concentrations of 20, 50, and 100 nM, there was a notable elevation in LDH leakage rates (*p* < 0.05), with the highest rate recorded at 100 nM (*p* < 0.05).

### 2.2. The Effect of T2 on ROS Activity, MDA Content, GSH Content and GPX Activity in DF-1 Cells

Figure 3 presents the data for the levels of ROS, MDA, GSH and the activity GPX in DF-1 cells. In comparison to the blank control group, oxidative stress caused by Nrf2 knockdown and T2 led to a significant increase in intracellular ROS, MDA content and GPX activity (*p* < 0.05), with the most pronounced effect observed when the two were combined (*p* < 0.05). Nrf2 overexpression resulted in a significant increase in all indicators except GPX activity (*p* < 0.05). Conversely, GSH content showed a significant decrease in all groups. When compared to the oxidative stress group, Nrf2 disruption resulted in a significant increase in the levels of ROS, MDA and GPX activity in cells (*p* < 0.05), but not GSH content. On the other hand, Nrf2 overexpression resulted in a significant decrease in the levels of ROS and GPX activity (*p* < 0.05), with no significant changes observed in other indicators (*p* > 0.05).

### 2.3. The Impact of T2 on the mRNA Expression of Factors Relevant to Oxidative Stress in DF-1 Cells

The impact of T2 on the mRNA expression of genes associated with the Nrf2-ARE oxidative stress signaling pathway was examined in DF-1 cells, as depicted in Figure 4. The results indicated that compared to the control group (0 nM), treatment of cells with 20 and 50 nM T2 led to a significant increase in the relative expression of Nrf2 (*p* < 0.05), with the most significant increase observed at 50 nM (*p* < 0.05). The relative expression of Keap1 in cells treated with 20, 50 and 100 nM T2 was found to be significantly reduced compared to the control group (*p* < 0.05). The relative expression of GPX1, NADPH: quinone oxidoreductase 1 (NQO1) and HO-1 was significantly increased in cells treated with T2 at concentrations of 20–100 nM (*p* < 0.05), with NQO1 showing the greatest increase in relative expression compared to the control group at 50 nM. No other significant differences were observed (*p* > 0.05).

### 2.4. Preparation and Formation of Recombinant Lentiviral Plasmids

PCR amplification was performed to obtain the target Nrf2 fragment, leading to the creation of the recombinant plasmid pLVML-Myc-Nrf2-IRES-Puro. The resulting samples were verified by double digestion and further confirmed through sequencing conducted by a company, which indicated that the eukaryotic expression vector overexpressing Nrf2 had been successfully constructed. Additionally, two sets of specific Nrf2 shRNA knockdown sequences, Nrf2shRNA-1 and Nrf2shRNA-2, were synthesized by Beijing Prime Tech Biotechnology Co., Ltd. The knockdown fragments were generated through consecutive annealing, with the sticky ends ligated to the pLKO.1-EGFP-Puro vector for transformation amplification. The outcome was the knockdown plasmids pLKO.1- Nrf2shRNA1-EGFP-Puro and pLKO.1-Nrf2shRNA2-EGFP-Puro, which were then sequenced by Beijing Prime Tech Biotechnology Co., Ltd. and were confirmed to have successfully constructed the lentiviral knockdown plasmid of Nrf2.

### 2.5. Assessment of the Impact of Modulated Nrf2 Expression

The results of the experiment showed that the DF-1 cells with knockdown Nrf2 expression were successfully obtained. The green fluorescence observed in the cells after 3–4 generations of drug selection using puromycin was due to the Enhanced Green Fluorescent Protein (EGFP) tag present in the lentiviral knockdown vector pLKO.1-EGFP-Puro (Figure 5A–C). The results showed that the knockdown with Nrf2 gene expression did not have a significant impact on the viability of DF-1 cells or on the expression of β-actin protein (*p* > 0.05) (Figure 5E), but it led to a significant reduction in mRNA levels (Figure 5D) and protein expression (Figure 5F,G) (*p* < 0.05). This indicates the successful construction of the Nrf2-knockdown cell line. Similarly, the overexpression of the Nrf2 gene did not have a significant effect on the cell viability (*p* > 0.05) but resulted in a significant increase in mRNA levels (Figure 5D) and protein expression (Figure 5F,G) compared to the control group (*p* < 0.05), demonstrating the successful construction of Nrf2 overexpression cell lines.

### 2.6. The Effect of Nrf2 Expression on the Morphology and Viability of DF-1 Cells

The cell morphology of DF-1 cells in each group after treatment with T2 (0 or 50 nM) for 24 h is depicted in Figure 6A–D. In comparison to the blank control group, the cells in the other three groups showed wider gaps, decreased numbers, and weakened connections between cells, resulting in an increased occurrence of unadhered cells and floating dead cells. The Nrf2-knockdown gene was found to have a more detrimental effect on cell morphology, causing a wider cell gap, a decrease in cell number, and an increase in dead and unadhered cells, as a result of oxidative stress. Conversely, Nrf2 overexpression led to a more tightly connected and increased number of cells, with fewer dead and unadhered cells.

In terms of cell activity, the results depicted in Figure 6E indicate a significant decrease (*p* < 0.05) compared to the blank control group in the other three groups, with the most significant decrease (*p* < 0.05) observed in the Nrf2-knockdown group. A comparison between the oxidative stress group and the Nrf2-knockdown group revealed a marked decrease (*p* < 0.05) in cell activity caused by the latter, while Nrf2 overexpression led to a significant increase (*p* < 0.05) in cell activity.

### 2.7. The Effect of Nrf2 Expression on ROS, MDA, GSH Content and GPX Activity in DF-1 Cells

The results of the measurement of ROS, MDA, GSH content, and GPX activity in DF-1 cells in each group are presented in Figure 7. The results demonstrate that compared to the control group, both Nrf2 knockdown and T2-induced oxidative stress result in a significant increase in ROS, MDA content, and GPX activity in the cells (*p* < 0.05), with the greatest impact observed when both were combined. Additionally, Nrf2 overexpression leads to a noticeable increase in ROS and MDA content (*p* < 0.05) and a pronounced decrease in GSH content in the cells (*p* < 0.05).

When compared to the T2 group, Nrf2 knockdown results in a significant increase in ROS, MDA content, and GPX activity in the cells (*p* < 0.05), along with a marked decrease in GSH content (*p* < 0.05). In contrast, Nrf2 overexpression causes a significant decrease in ROS content and GPX activity (*p* < 0.05), while the changes in MDA and GSH content were not statistically significant (*p* > 0.05).

### 2.8. The Effect of Nrf2 Expression on the Expression of Oxidative Stress-Related Factor mRNA

Figure 8 presents the mRNA expression data of oxidative stress-related factors for each group. In comparison to the control group, both T2-induced oxidative stress and Nrf2 overexpression led to a substantial elevation in the relative expression of Nrf2, GPX1, NQO1, and HO-1 in the cells (*p* < 0.05), with the greatest impact observed when both conditions were present (*p* < 0.05). The relative expression of Keap1 mRNA in the cells of each group was significantly lower. With respect to the T2 group, Nrf2 overexpression resulted in a marked increase in the relative expression of Nrf2 and GPX1 in cells (*p* < 0.05), while Nrf2 knockdown caused a pronounced decrease in the relative expression of Nrf2, GPX1, NQO1, and HO-1 in cells (*p* < 0.05).

### 2.9. The Effect of Nrf2 Expression on the Expression of Oxidative Stress-Related Proteins

Figure 9 displays the results of the analysis of protein expression levels of oxidative stress-related factors in each group of DF-1 cells. Compared to the blank control group, Nrf2 overexpression and oxidative stress induced by T2 treatment both elicited a marked increase in the relative expression of Nrf2, GPX1, and HO-1 proteins in the cells (*p* < 0.05). On the other hand, Nrf2 knockdown gene alone resulted in a notable reduction in the relative expression of HO-1 protein in the cells (*p* < 0.05). When comparing the oxidative stress group to Nrf2 overexpression group, it was observed that the latter group showed a remarkable enhancement in the relative expression of GPX1 protein in the cells (*p* < 0.05). The Nrf2 knockdown gene led to a substantial decrease in the relative expression of Nrf2, GPX1, and HO-1 proteins in the cells, with statistical significance (*p* < 0.05).

## 3. Discussion

T2, classified as a class A mono-enriched mycotoxin, stands out as the most toxic among its peers. Notably, T2 is acknowledged as a significant contaminant in animal feed, posing a potential threat to human health due to its persistence in animal-derived food products [14,16]. Consequently, considerable research has been dedicated to comprehending its cytotoxicity and its role in inducing oxidative stress [23]. T2 has been recognized as an instigator of cellular oxidative stress, a pivotal determinant of cellular damage. The Nrf2-ARE signaling pathway has been established as a crucial mechanism for cellular antioxidant regulation, with nuclear factor E2-related factor 2 (Nrf2) emerging as a key player in this context [24]. However, there has been a gap in our knowledge concerning whether Nrf2 regulates T2-induced oxidative stress. Therefore, the primary objective of this study was to explore the influence of the Nrf2 protein, derived from broilers, on the regulation of oxidative damage induced by T2 [1,2].

T2 exhibits significant cytotoxicity. The outcomes of the present investigation revealed that exposure to T2 led to a reduction in DF-1 cell viability and an increase in intracellular LDH leakage. These findings align with prior research conducted by Weidner et al. [25]. T2 is capable of inhibiting protein and nucleic acid synthesis, disrupting cell cycle progression, provoking oxidative stress, and inciting apoptosis and necrosis in various animal organs, including the liver, kidneys, and the immune system [21,26]. The heightened intracellular LDH leakage serves as an indicator of T2’s ability to compromise cell membrane integrity, thus affirming its cytotoxic properties [27]. Additionally, the observed decrease in cell count and morphological alterations under microscopic examination provide concrete evidence of T2’s detrimental impact on DF-1 cells.

T2 induces oxidative stress in cells, as evidenced by the increase in ROS and MDA content and the decrease in GSH content, accompanied by heightened GPX activity in DF-1 cells following T2 exposure. These results align with the research conducted by Yang et al. [28]. Notably, our investigation unveiled that the intracellular MDA content exhibited a dose-dependent increase within the T2 concentration range of 0–50 nM. However, when the T2 concentration reached 100 nM, the MDA content was observed to be lower (*p* < 0.05). This anomaly can be attributed to the substantial cell death induced by the high T2 concentration of 100 nM, which consequently led to an undetectable intracellular MDA content. In the early stages of oxidative stress, intracellular GSH levels tend to rise. Yet, when these levels are inadequate to counter intracellular oxidative stress, it results in an escalation of intracellular ROS levels, subsequently contributing to cellular damage. Indeed, our study established that T2 treatment significantly elevated intracellular ROS levels, indicating that ROS serves as one of the factors inducing cell death [12].

Furthermore, our investigation demonstrated a considerable increase in intracellular GPX activity due to T2 exposure, with the activities of antioxidant enzymes exhibiting dose-dependent elevation in response to T2. GSH, as a vital disulfide redox buffer, plays a paramount role in preserving cells within a reducing environment during redox reactions [28,29]. However, when exposed to excessively high T2 doses (100 nM), cells may fail to produce an adequate amount of GSH to counteract oxidative damage, leading to a decrease in GSH levels [12]. Nevertheless, GSH retains its significance as a pivotal antioxidant in the cell. Additionally, the presence of GSH acts as a substrate for enhancing intracellular antioxidant enzymes and maintaining cellular homeostasis. Subsequent to the occurrence of oxidative stress in a cell, the corresponding antioxidant enzymes are generated within the cell to combat the damage induced by oxidative stress [30,31]).

In this experiment, we observed a dose-dependent increase in the relative mRNA expression of GPX1, Nrf2, O-1, and NQO1 genes in response to T2, a pattern congruent with the results obtained for intracellular MDA. Nrf2 holds a pivotal role in the Nrf2-ARE signaling pathway, a vital defense mechanism against oxidative damage and a central signaling pathway responsible for enhancing the organism’s antioxidant capacity. This pathway is implicated in the cellular oxidative stress response, exerting its antioxidant effects by regulating the downstream activity of antioxidant enzymes [32]. Our study unveiled that T2 induced the upregulation of Nrf2, which subsequently interacted with ARE in the nucleus, initiating the expression of downstream antioxidant enzyme genes, such as GPX1, NQO1, and HO-1. This process bolstered cellular resilience against oxidative damage, enabling the removal of ROS and other detrimental substances [33,34]. Notably, at 100 nM, the expression of most oxidative stress-related mRNAs (Nrf2, NQO1, GPX1) was lower than that at 50 nM, a phenomenon that aligns with the increased cell death observed due to high T2 doses (100 nM).

The 293FT cells were utilized for lentiviral (LV) vector packaging instead of the commonly used 293T cells. This choice was motivated by the inherent possession of SV40 T antigen in 293FT cells, which holds the potential to enhance the viral titer of LV and concurrently mitigate the activation of the intracellular innate immune response [35,36]. The use of shRNA, a mechanism for specific gene silencing, was employed to stabilize the expression of target genes through the in vitro synthesis of targeted interfering shRNA sequences [37]. The findings affirmed that the amplified broiler-derived Nrf2 was successfully ligated to the overexpression lentiviral vector pLVML-Myc-MCS-IRES-Puro, resulting in the construction of the Nrf2 overexpression-DF-1 cell line (DF-1-Nrf2). This construction was validated through real-time PCR and Western blot analysis, which conclusively disclosed a substantial elevation in Nrf2 mRNA and protein levels within DF-1-Nrf2 cells (*p* < 0.05). Although fluorescence microscopy indicated the presence of green fluorescence in the DF-1-Nrf2KD1 and DF-1-Nrf2KD2 cell lines and empty carrier cells, the fluorescence rate was less than 100% due to resistance to puromycin in some DF-1 cells. The results from the MTT assay showed no significant differences in absorbance values between DF-1-Nrf2, DF-1-Nrf2KD1, and DF-1-Nrf2KD2 cell lines. This unequivocally indicated the maintenance of normal cellular activities within these lines. This study thus presents a method for the stable overexpression and knockdown of Nrf2 in DF-1 cell lines, providing a valuable tool for future studies of Nrf2-related signaling pathways and facilitating clinical research of Nrf2.

The authors of this study initially formulated the hypothesis that the Nrf2-ARE signaling pathway orchestrates the modulation of T2-induced oxidative stress. This hypothesis was subsequently fortified through the lens of previous research and scholarly works. The ARE signaling pathway was found to have antioxidant properties by uncoupling Nrf2 from Keap1 and interacting with ARE, thus activating downstream antioxidant enzymes [38,39]. It is speculated that the Nrf2-ARE pathway plays a crucial role in the T2-induced oxidative stress process.

In DF-1 cells, Nrf2, a central factor, plays a pivotal role in modulating the extent of cellular damage induced by T2. The data from our study demonstrated that a 24 h treatment with 50 nM of T2 resulted in a notable reduction in the number of DF-1 cells displaying aberrant morphology and a significant decline in cellular viability when Nrf2 expression was interfered with (*p* < 0.05). Conversely, the overexpression of Nrf2 prompted an increase in the number of DF-1 cells, a reduction in the extent of morphological damage, and a substantial enhancement in cellular viability (*p* < 0.05).

In the results of oxidative stress-related indexes, the treatment of 50 nM T2 for 24 h, interfering with the expression of Nrf2 led to a significant up-regulation of ROS content, MDA content and GPX activity (*p* < 0.05) and a significant decrease in the concentration of GSH in DF-1 cells (*p* < 0.05), while the overexpression of Nrf2, on the contrary, caused a significant down-regulation of the ROS content and the activity of GPX (*p* < 0.05). The results can preliminarily prove that in DF-1 cells, Nrf2, a key factor, can regulate the oxidative stress in DF-1 cells, and up-regulation of Nrf2 can reduce the level of cellular oxidative stress, while down-regulation of Nrf2 can increase the level of cellular oxidative stress.

In the results of Nrf2-ARE signaling pathway-related factors, the mRNA and protein expression of Nrf2 were as expected. After 24 h of treatment with 50 nM T2, the mRNA and protein levels of Nrf2 in Nrf2 overexpressing-DF-1 cells were significantly increased (*p* < 0.05), and the mRNA and protein levels of Nrf2 interfering-DF-1 cells were significantly decreased (*p* < 0.05). In contrast, there was no significant difference in Keap1 mRNA expression in both Nrf2 overexpressed-DF-1 cells and Nrf2 interfered-DF-1 cells (*p* > 0.05), which suggests that the interference and overexpression of Nrf2 had no significant effect on Keap1 expression in the Nrf2-ARE signaling pathway. In addition, in the results of the assay of related factors of antioxidant enzyme lineages downstream of Nrf2-ARE signaling pathway, it was found that the mRNA levels of GPX1, HO-1 and NQO1 and the protein expression levels of GPX1 and HO-1 were significantly down-regulated in Nrf2-disturbed-DF-1 cells (*p* < 0.05); and the mRNA levels of GPX1, HO-1 and NQO1 and the protein expression levels of GPX1 and HO-1 were significantly down-regulated in Nrf2-overexpressed-DF-1 cells under the same treatment conditions. In Nrf2 overexpression-DF-1 cells, the GPX1 mRNA and protein expression were significantly up-regulated (*p* < 0.05). Therefore, the above results can confirm that in DF-1 cells, the level of cellular oxidative stress can be regulated by up-regulating and down-regulating Nrf2 protein expression, which in turn regulates the expression of antioxidant enzymes downstream of the Nrf2-ARE signaling pathway, and enhances/decreases the activity of antioxidant enzymes.

Despite alterations in Nrf2 expression and the antioxidant enzyme cascade, no significant changes were observed in MDA content or GSH concentration in the Nrf2 overexpression group of DF-1 cells following treatment with 50 nM T2. Similarly, variations in factors linked to the Nrf2-ARE signaling pathway, such as NQO1, HO-1, and HO-1, did not reach statistical significance. These observations may be attributed to potential inefficiencies in the construction of DF-1-Nrf2 cells for overexpression.

The results of the present study emphasize the critical role of Nrf2 in regulating T2-induced oxidative stress in DF-1 cells, mainly through the activation of antioxidant enzymes downstream of the Nrf2ARE signaling pathway to exert antioxidant activity. This provides a new idea for the subsequent treatment of oxidative stress-related diseases in livestock and poultry, and Nrf2 can be used as an effective target of drug action to treat oxidative stress-related diseases in livestock and poultry by overexpression of the Nrf2 side.

In the analysis of oxidative stress-related parameters, exposure to 50 nM T2 for 24 h resulted in a noteworthy up-regulation of ROS content, MDA levels, and GPX activity in DF-1 cells when Nrf2 expression was disrupted, signifying significant differences (*p* < 0.05). Furthermore, this interference led to a marked decrease in cellular GSH concentrations (*p* < 0.05). Conversely, the overexpression of Nrf2 had the opposite effect, significantly reducing ROS content and GPX activity (*p* < 0.05). These outcomes offer preliminary evidence that Nrf2, a pivotal factor in DF-1 cells, can modulate cellular oxidative stress. Up-regulation of Nrf2 reduces oxidative stress, while its down-regulation increases oxidative stress levels.

In the context of the Nrf2-ARE signaling pathway-related elements, the expression of Nrf2 exhibited anticipated trends. Following 24 h exposure to 50 nM T2, Nrf2 overexpressing-DF-1 cells displayed a substantial increase in both mRNA and protein levels of Nrf2 (*p* < 0.05), whereas Nrf2-interfered DF-1 cells exhibited a marked decrease (*p* < 0.05). However, Keap1 mRNA expression remained insignificantly affected in both the Nrf2-overexpressing and Nrf2-interfered DF-1 cells (*p* > 0.05), indicating that Nrf2 interference and overexpression had a negligible impact on Keap1 expression within the Nrf2-ARE signaling pathway. Moreover, concerning the evaluation of antioxidant enzyme lineage factors downstream of the Nrf2-ARE signaling pathway, the mRNA levels of GPX1, Hmox1, and Noq1, alongside the protein expression levels of GPX1 and HO-1, were significantly down-regulated in Nrf2-interfered DF-1 cells (*p* < 0.05). Paralleling these findings, Nrf2-overexpressed DF-1 cells under the same treatment conditions exhibited a substantial down-regulation in the mRNA levels of GPX1, Hmox1, and Noq1, as well as the protein expression levels of GPX1 and HO-1 (*p* < 0.05). However, in the Nrf2 overexpression group, GPX1 mRNA and protein expression were notably up-regulated (*p* < 0.05). These results conclusively illustrate that Nrf2 protein expression, when upregulated or downregulated, has the capacity to govern the level of oxidative stress in DF-1 cells. This regulation extends to influencing the expression of antioxidant enzymes downstream of the Nrf2-ARE signaling pathway and subsequently modulating the activities of these enzymes.

It is worth noting that despite variations in Nrf2 expression and the antioxidant enzyme cascade, no significant changes were observed in the MDA content or GSH concentration within the Nrf2 overexpression group of DF-1 cells after exposure to 50 nM T2. Similarly, modifications in factors related to the Nrf2-ARE signaling pathway, such as NQO1, HO-1, and HO-1, did not reach statistical significance. These observations may be attributed to potential inefficiencies in the construction of DF-1-Nrf2 cells for overexpression.

The findings from this study underscore the pivotal role of Nrf2 in modulating T2-induced oxidative stress in DF-1 cells. This regulation predominantly occurs through the activation of downstream antioxidant enzymes within the Nrf2-ARE signaling pathway, contributing to overall antioxidant activity. These insights provide a novel perspective for the subsequent treatment of oxidative stress-related conditions in livestock and poultry. Nrf2 may serve as an effective target for therapeutic intervention in oxidative stress-related ailments in livestock and poultry through Nrf2 overexpression.

The study established that T2’s toxic impact on DF-1 cells and ensuing oxidative stress amplified with increasing T2 doses. The pivotal role of the Nrf2-ARE signaling pathway was confirmed. To delve into Nrf2’s function, stable Nrf2 overexpression or suppression in DF-1 cell lines was achieved via lentiviral expression, effectively regulating Nrf2 without impeding cellular activity. These outcomes strongly underscore Nrf2’s indispensable role in governing T2-induced oxidative stress in DF-1 cells, achieved through modulating Nrf2 expression via the Nrf2-ARE signaling pathway.

## 4. Conclusions

This study was undertaken to scrutinize the detrimental impact of T2 on DF-1 cells and dissect its molecular engagement in oxidative stress. This research lays the groundwork for further explorations into the noxious consequences of this toxin on broilers. Moreover, it has unveiled the functional role of Nrf2 in ameliorating T2-induced oxidative stress, shedding light on the underlying mechanisms. Significantly, this study underscores the potential of Nrf2 within the Keap1-Nrf2/ARE pathway as a robust drug target, thus broadening the horizons for therapeutic interventions in managing oxidative stress-related maladies in livestock. Concurrently, the study has both established and refined the methodological framework for achieving Nrf2 gene overexpression and interference in DF-1 cells.

## 5. Materials and Methods

### 5.1. Chemicals and Reagents

T2 was purchased from Pribolab Bioengineering Co., Ltd. (Qingdao, Shandong, China). Dulbecco’s modified Eagle medium (DMEM), fetal bovine serum (FBS) and penicillin-streptomycin double antibody solution (P/S) were obtained from Thermo Fisher Scientific (Waltham, MA, USA). Four lentiviral plasmids (psPAX2, pMD2.G, pLKO.1-EGFP-Puro, and pLVML-Myc-MCS-IRES-Puro) were purchased from Miao Ling Biotechnology Co., Ltd. (Wuhan, Hubei, China). Recombinant glutathione peroxidase 1 (GPX1) antibody and HO-1 antibody were provided by Bioss Biotechnology Co., Ltd. (Beijing, China).

### 5.2. Kits

The lipid oxidation (MDA), glutathione (GSH), glutathione peroxidase (GPX) and lactate dehydrogenase cytotoxicity (LDH) assay kits were purchased from Beyotime Biotechnology (Shanghai, China). The ROS assay kit was purchased from US EVERBRIGHT INC (Silicon Valley, CA, USA). The M-MLV First Strand cDNA synthesis kit was obtained from Gene Copoeia Inc. (Rockville, MD, USA). The Bicinchoninic acid (BCA) protein assay kit was obtained from Thermo Fisher Scientific (Waltham, MA, USA).

### 5.3. Cell Culture

The Chicken embryo fibroblast cells (DF-1) utilized in this study were purchased from Fenghui Biotechnology Co., Ltd. (Changsha, Hunan, China). The cells were maintained in a 10% complete medium; when the cells were fully covered to 90%, they were continuously passaged in a 1:3 ratio [40].

### 5.4. The MTT (3-[4,5-dimethylthiazol-2-yl]-2,5 Diphenyl Tetrazolium Bromide) Assay for Cell Activity and IC_50_ Value

The DF-1 cells were grown in 96-well plates for 24 h. The cells were then treated with varying concentrations of T2 (0, 10, 20, 50, 100 nM) in separate groups. After 24 and 48 h, MTT solution was added to each well and incubated for 4–6 h [13,41,42]. The MTT solution was then removed, and the reaction was stopped by adding dimethyl sulfoxide (DMSO) to each well and stirring for 10 min to dissolve the formed crystals. The absorption values were measured at 490 nm using an ELISA reader and the results were analyzed [43]. The T2 concentration required to halve the absorbance value of the control group is the IC_50_ concentration [44]. These procedures were performed in a sterile environment.

### 5.5. Measurement of Intracellular GSH, LDH, ROS, MDA and GPX

After the 6-well plate cells adhered to the wall, we inoculated DF-1 cells into the 96-well plate and allowed them to grow for 24 h. Afterward, 100 μL of T2 (0, 10, 20, 50, 100 nM) was added to the cells, which were grouped accordingly. At 24 and 48 h post-treatment, various indexes were determined, including the LDH leakage rates, ROS, MDA content levels, GSH levels, and GPX activities, in accordance with the manufacturer’s instructions [40,45,46,47,48].

### 5.6. Relative mRNA Levels of Oxidative Stress-Related Genes in DF-1 Cells

The process involved extracting total RNA from cells using Trizol reagent. The RNA concentration was measured, ensuring that selected RNA samples had purity ratios (A260/A280) within the range of 1.9–2.0 and A260/A230 greater than 2.0, with a stand-ardized concentration of 0.5 ng/mL. Subsequently, the RNA was reverse transcribed into cDNA using the M-MuLV first strand cDNA synthesis kit. Table 1 lists the primer sequences for the relevant genes in this experiment. The resulting cDNA served as a template for the quantitative PCR reaction, with the reaction system detailed in Table 2. The reaction conditions comprised an initial step at 95 °C for 10 min, followed by 39 cycles of 95 °C for 10 s and 60 °C for 30 s. A melting curve was generated by running the reaction at 95 °C for 15 s, 60 °C for 1 min, and 95 °C for 15 s. The relative mRNA expression in the test group was determined using the 2^−ΔΔCt^ method, which involved comparing the Ct value of the test group gene with that of the internal reference gene. Each PCR reaction was carried out in triplicate [49]

### 5.7. Preparation of Recombinant Lentiviral Plasmid

The total genome of DF-1 cells served as the template for amplification. Using specific primers and general purification, the full-length Nrf2 gene was amplified. The pLKO.1-EGFP-Puro, Nrf2 full-length and pLVML-Myc-MCS-IRES-Puro vectors were digested with restriction endonucleases (Age I and Eco RI or Eco RI and Bam HI) and subsequently purified through gel extraction. The purified vector was then ligated with the shRNA sequences using T4 ligase and landing PCR. The resulting recombinant plasmid was introduced into Escherichia coli DH5α recipient cells, with monoclonal colonies being cultured at 37 °C for 16 h. Subsequently, these colonies were transferred to ampicillin (AMP)-resistant Luria–Bertani (LB) medium and shaken at 37 °C for 15 h [50]. The amplified colonies were subjected to small extraction of the plasmids using the Plasmid Small Volume Rapid Extraction Kit (Plasmid Small Extraction Kit). Then, we prepare 1% gel and electrophoresis equipment to take 1 μL of the recombinant plasmid for the electrophoresis experiment. At the end of electrophoresis, the gel block was placed into the gel imager and the electrophoresis results were analyzed. The positive recombinant plasmid pLVML-Myc-Nrf2-IRES-Puro, confirmed through gel verification, was further verified by double digestion with EcoRI and BamHI. To achieve this, a sterile centrifuge tube was prepared, containing 6 μL of sterile water, 1 μL of 10× enzyme digestion buffer, 0.5 μL of EcoRI, 0.5 μL of BamHI, and 2 μL of the recombinant plasmid. The mixture was gently mixed, left briefly, and incubated at 37 °C for 45–60 min for enzyme digestion. At the end of the process, electrophoresis was performed, and the results were analyzed in a gel imager. The positively verified recombinant plasmid underwent sequencing to confirm sequence homology and correctness (Beijing Tsingke Biotech Co., Ltd. Beijing, China).

The Human Embryonic Kidney Cells (293FT), preserved in the laboratory, were inoculated into cell culture dishes, grown in DMEM medium with 10% FBS in a 37 °C incubator with 5% CO2, and the medium was changed to DMEM without FBS when the cell density reached approximately 80%. A mixture containing 10 μg of the recombinant lentiviral plasmid and empty vector and 5 μg of the helper plasmids (psPAX2 and pMD2.G) was fixed with Opti-MEM and combined with Lipofectamine 2000 (Lip2000). After incubation for 6 h and replacement with DMEM medium containing 2% FBS for another 48 h, the supernatant was collected by repeated freeze–thaw cycles and filtered through a 0.22 μm filter before storage at −80 °C.

### 5.8. Construction of Nrf2 Knockdown and Overexpression in DF-1 Cell Line

The DF-1 cell line was inoculated into 6-well plates and incubated in DMEM medium containing 10% fetal bovine serum (FBS) at 37 °C and 5% CO_2_ until the cell density reached approximately 70%. The medium was then aspirated and replaced with a fresh DMEM medium containing 2% FBS, followed by the introduction of recombinant lentivirus (pLVML-Myc-Nrf2-IRES-Puro, pLKO.1-Nrf2shRNA1- EGFP-Puro, pLKO.1-Nrf2shRNA2-EGFP-Puro) or an empty vector. Following 6 h incubation, when the cell density approached 100%, puromycin (2 μg/mL) was added for cell selection. Surviving cells were re-plated into 96-well plates and cultured for an additional 48 h in DMEM medium containing 2% FBS [51,52]. Monoclonal cell colonies were subsequently selected for amplification and subjected to verification using various techniques, including microscopy, MTT assay, real-time PCR, and Western blot analysis.

### 5.9. Assessment of the Impact of Modulated Nrf2 Expression

The DF-1-Nrf2KD1 cells, DF-1-Nrf2 cells, and DF-1 cells were grouped into four categories: blank control (normal DF-1 cells without T2 treatment), T2, T2 + Nrf2 (+) (T2 + Nrf2 overexpression), T2 + Nrf2 (−) (T2 + Nrf2 knockdown) for cultivation. After 24 h of incubation, the media were changed based on the grouping, with the addition of 50 nM T2. After a further 24 h of incubation, the cells were collected and analyzed through cell morphology, MTT assay, Western blotting, real-time PCR, and measurement of intracellular ROS levels, MDA content, GSH content, and GPX activity.

### 5.10. Statistical Analysis

The entire design process of this experiment is shown in Figure 10. The statistical analysis of the experimental data was conducted using the SPSS Statistics 26.0 software from SPSS Inc. (Chicago, IL, USA). The results of the IC_50_ values and the fluorescent quantitative PCR tests were presented graphically using GraphPad Prism 8 (GraphPadSoftware San Diego, CA, USA). One-way analysis of variance (ANOVA) was performed to compare all groups and pairwise comparisons were performed using the Tukey test. tests were repeated three times, and test results are expressed as mean ± standard error. A significance level of *p* < 0.05 was used to indicate a significant difference.

## Figures and Tables

**Figure 1 toxins-15-00627-f001:**
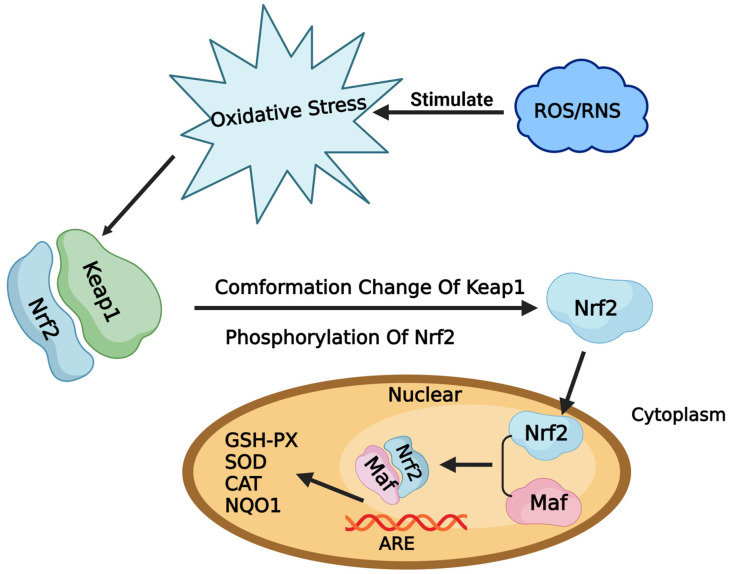
Nrf2-keap1/ARE signaling pathway.

**Figure 2 toxins-15-00627-f002:**
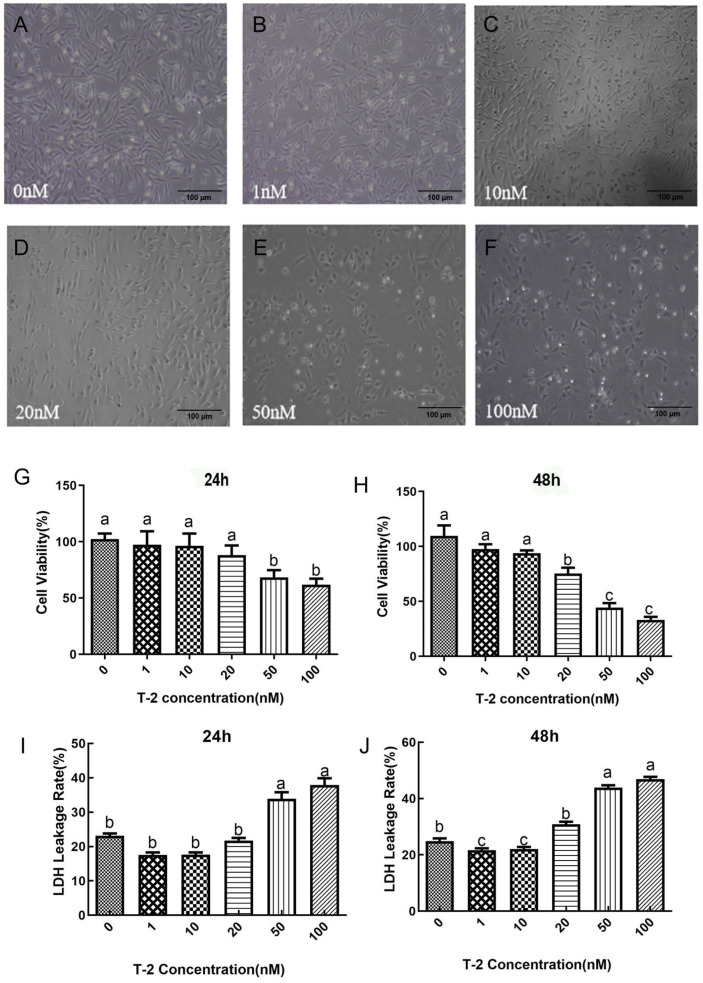
Microscopic observation in chicken embryo fibroblast cell line (DF-1) cells in response to varying concentrations of T-2 toxin (T2) treatment (100×). Micrograph of control cells (**A**); micrograph of 1 nM T2-treated cells (**B**); micrograph of 10 nM T2-treated cells (**C**); micrograph of 20 nM T2-treated cells (**D**); micrograph of 50 nM T2-treated cells (**E**); micrograph of 100 nM T2-treated cells (**F**). The cell viability (%) of DF-1 cell treated with the different concentrations of T2 for 24 h (**G**) or 48 h (**H**). The lacate dehydrogenase (LDH) leakage rate (%) of DF-1 cell with the different concentrations of T2 for 24 h (**I**) and 48 h (**J**). Note: Values with the difference superscript letters (a–c) are significant differences (*p* < 0.05).

**Figure 3 toxins-15-00627-f003:**
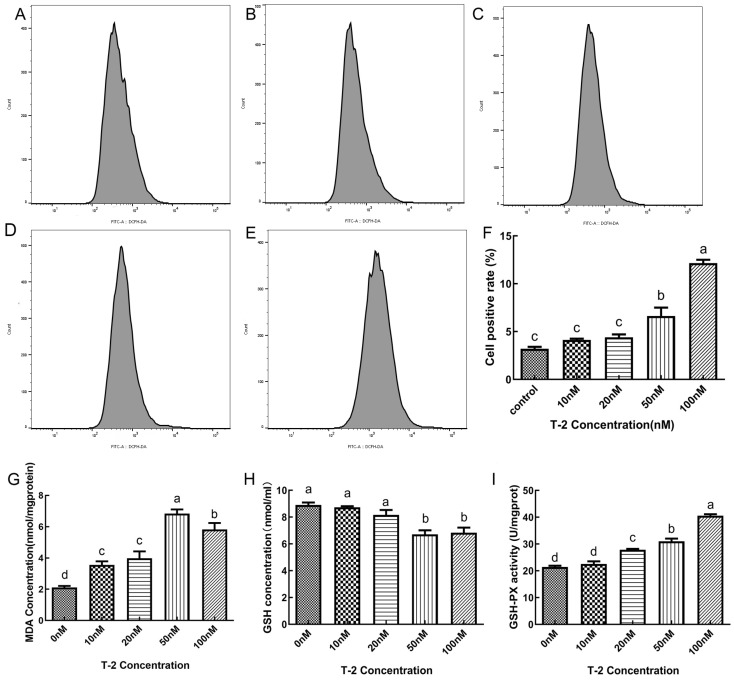
Reactive oxygen species (ROS) production in T2-treated or untreated DF-1 cells for 24 h. Representative images of peak ROS-positive cells treated with T2 at concentrations of 0 nM (**A**), 10 nM (**B**), 20 nM (**C**), 50 nM (**D**), and 100 nM (**E**); histogram of ROS-positive cells were quantified (**F**). Comparison of malondialdehyde (MDA) concentration (nmol-mgprotein^−1^) in T2-treated or untreated DF-1 cells (**G**); comparison of glutathione (GSH) concentration (nmol-mL^−1^) in T2-treated or untreated DF-1 cells (**H**); comparison of glutathione peroxidase (GSH-PX) activity (U-mgprot^−1^) in T2-treated or untreated DF-1 cells (**I**). Note: Values with the difference superscript letters (a–d) are significant differences (*p* < 0.05).

**Figure 4 toxins-15-00627-f004:**
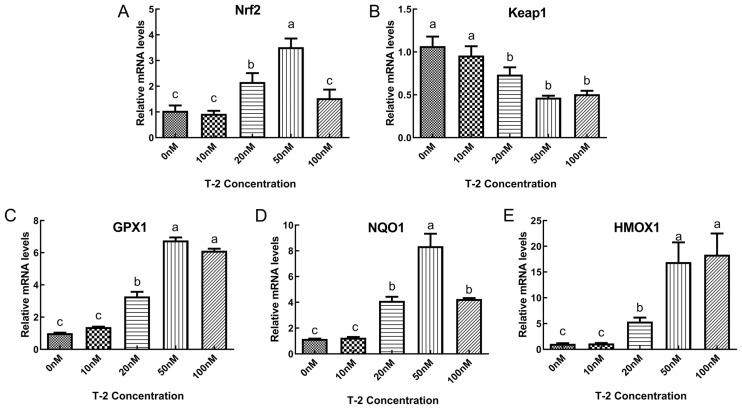
The relative expression levels of nuclear factor E2 related factor 2 (Nrf2) (**A**), kelch-like ECH-associated protein 1 (Keap1) (**B**), recombinant glutathione peroxidase1 (GPX1) (**C**), NADPH: quinone oxidoreductase 1 (NQO1) (**D**) and heme oxygenase 1 (HO-1) (**E**) mRNAs in T2-treated or untreated DF-1 cells. Note: Values with the difference superscript letters (a–c) are significant differences (*p* < 0.05).

**Figure 5 toxins-15-00627-f005:**
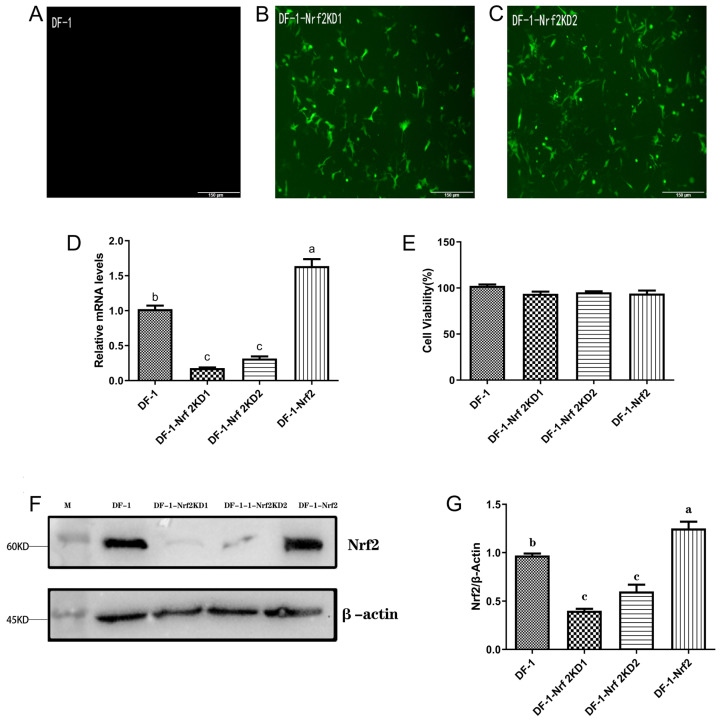
Microscopic observation of Enhanced Green Fluorescent Protein (EGFP) expression in normal DF-1 cells (**A**) and Nrf2 knockdown DF-1 cells (**B**,**C**) (200×), the EGFP label can exhibit green fluorescence when observed under a fluorescence microscope. Nrf2 mRNA expression in knockdown and overexpression cell lines (**D**). Cell viability (%) of Nrf2 knockdown and overexpression cell lines (**E**). Immunoblot analysis of Nrf2 protein in disturbed and overexpressed cell lines (**F**); relative expression data of Nrf2 protein in disturbed and overexpressed cell lines (**G**). Note: M:26619 Marker. Values with the difference superscript letters (a–c) are significant differences (*p* < 0.05).

**Figure 6 toxins-15-00627-f006:**
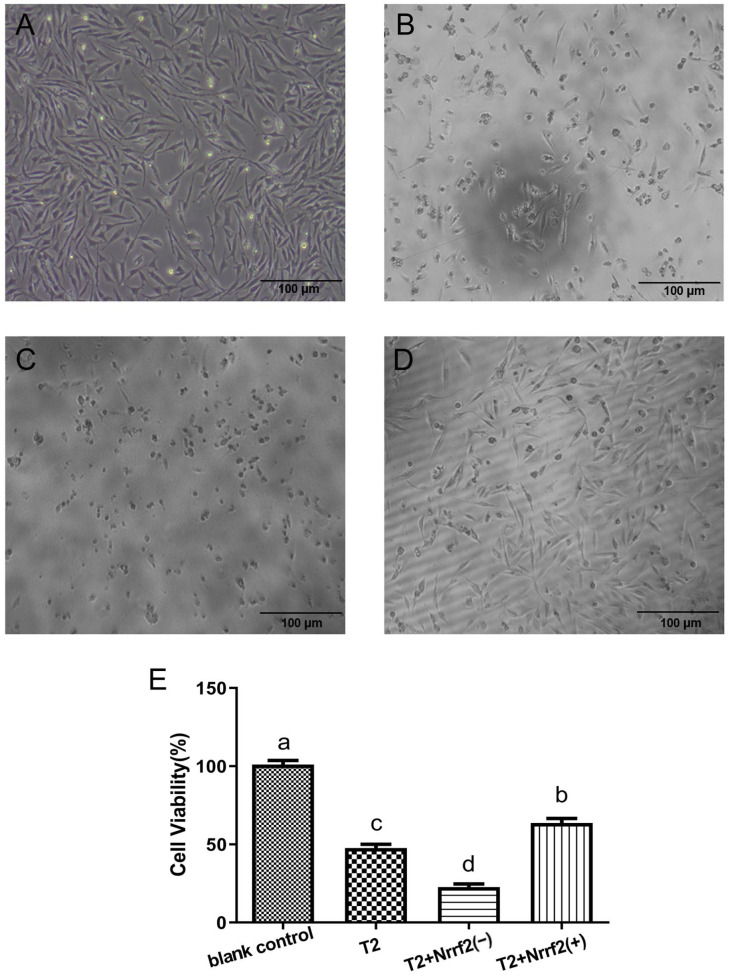
Microscopic observation of DF-1 cells in response to changes in Nrf2 expression (100×). Micrograph of blank control group cells (**A**); micrograph of T2 group cells (**B**); micrograph of T2 + Nrf2(−) group cells (**C**); micrograph of T2 + Nrf2(+) group cells (**D**). The cell viability (%) of DF-1 cells upon changes in Nrf2 expression (**E**). Note: Values with the difference superscript letters (a–d) are significant differences (*p* < 0.05).

**Figure 7 toxins-15-00627-f007:**
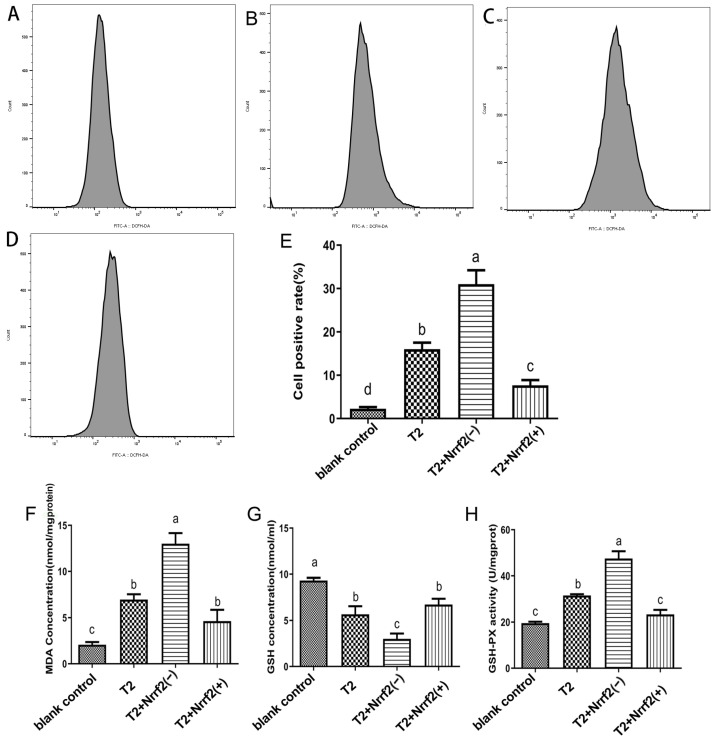
Reactive oxygen species (ROS) production in T2-treated or untreated normal DF-1 cells, Nrf2 knockdown DF-1 cells and overexpressed DF-1 cells for 24 h. Representative images of peak ROS-positive cells in the blank control (**A**), T2 (**B**), T2 + Nrf2 (−) (**C**) and T2 + Nrrf2(+) (**D**) groups; histogram of ROS-positive cells was quantified (**E**). Comparison of MDA concentration (nmol-mgprotein^−1^) (**F**), GSH concentration (nmol-mL^−1^) (**G**) and GPX viability (U-mgprot^−1^) (**H**) in T2-treated or untreated normal DF-1 cells, Nrf2 knockdown DF-1 cells and overexpressed DF-1 cells for 24 h. Note: Values with the difference superscript letters (a–d) are significant differences (*p* < 0.05).

**Figure 8 toxins-15-00627-f008:**
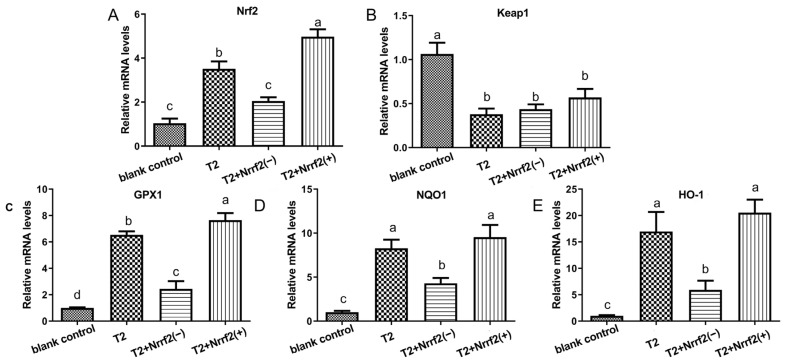
The relative expression levels of Nrf2 (**A**), Keap1 (**B**), GPX1 (**C**), NQO1 (**D**) and HO-1 (**E**) mRNAs in T2-treated or untreated normal DF-1 cells, Nrf2 knockdown DF-1 cells and overexpressed DF-1 cells. Note: Values with the difference superscript letters (a–d) are significant differences (*p* < 0.05).

**Figure 9 toxins-15-00627-f009:**
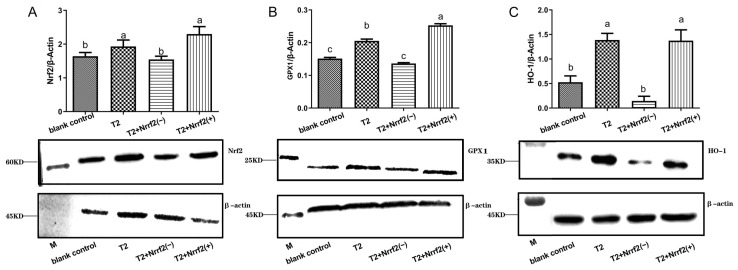
Relative expression data and protein immunoblot analysis of oxidative stress-related factors Nrf2 (**A**), GPX1 (**B**) and HO-1 (**C**) proteins in T2-treated or untreated normal DF-1 cells, Nrf2 knockdown DF-1 cells and overexpressed DF-1 cells for 24 h. Note: Values with the difference superscript letters (a–c) are significant differences (*p* < 0.05).

**Figure 10 toxins-15-00627-f010:**
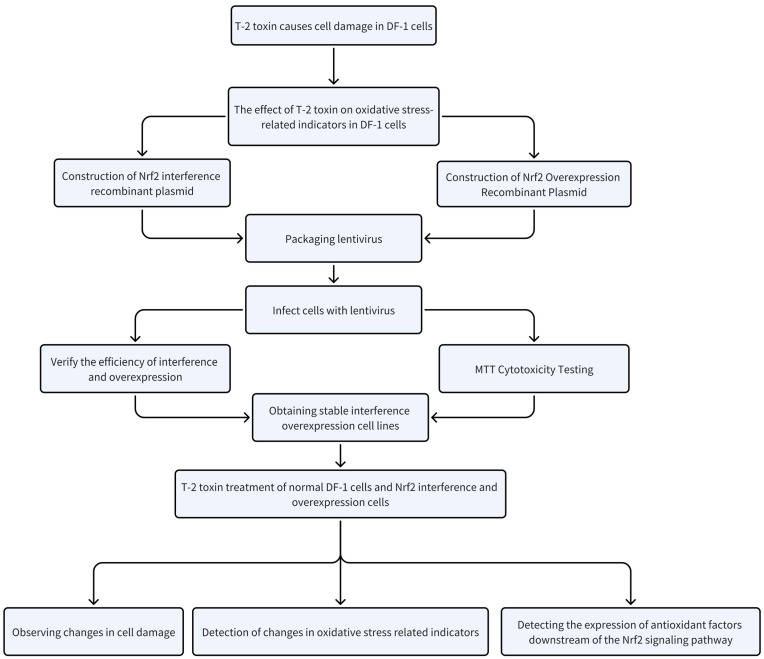
Experimental design flow chart.

**Table 1 toxins-15-00627-t001:** Information of primers/sequence.

Primer/Sequence Name	Primer Sequence (5′→3′)	GenBank Accession No.
Nrf2-F	CATAGAGCAAGTTTGGGAAGAG	MN416129.1
Nrf2-R	GTTTCAGGGCTCGTGATTGT	MN416129.1
Keap1-F	ACTTCGCTGAGGTCTCCAAG	MN416132.1
Keap1-R	CAGTCGTACTGCACCCAGTT	MN416132.1
GPX1-F	CTGTTCGCCTTCCTGAGAGA	NM_001277853.1
GPX1-R	TGCAGTTTGATGGTCTCGAA	NM_001277853.1
HO-1-F	AGCGCAGCGCTTCACGTCCC	NM_205344.2
HO-1-R	ATAAAAGTCAATGTAAAGCG	NM_205344.2
NQO1-F	AAGCCCTGCCCCGAGCGAAG	NM_001277621.1
NQO1-R	GCAACAATAAATCGAGGTCT	NM_001277621.1
GAPDH-F	GAACATCATCCCAGCGTCCA	NM_204305.1
GAPDH-R	CGGCAGGTCAGGTCAACAAC	NM_204305.1
Nrf2-MYC-F	CCGGAATTCATGGAGCAGAAACTCATCTCTGAAGAGGATCTGAACTTGATTGACATC	MN416129.1
Nrf2-Ha-R	CGCACCGGTTCAAGCGTAATCTGGAACATCGTATGGGTACAGTTTAGTCTCTGCCTT	MN416129.1
Nrf2-shRNA-F-1	CCGGGGAGACAGGTGAATTTGTTCCTCAAGAGGGAACAAATTCACCTGTCTCCTTTTT	MN416129.1
Nrf2-shRNA-R-1	AATTAAAAAGGAGACAGGTGAATTTGTTCCCTCTTGAGGAACAAATTCACCTGTCTCC	MN416129.1
Nrf2-shRNA-F-2	CCGGGCAAGTTTGGGAAGAGTTATTTCAAGAGAATAACTCTTCCCAAACTTGCTTTTT	MN416129.1
Nrf2-shRNA-R-2	AATTAAAAAGCAAGTTTGGGAAGAGTTATTCTCTTGAAATAACTCTTCCCAAACTTGC	MN416129.1

**Table 2 toxins-15-00627-t002:** Real-time quantitative PCR system.

Reagent	Dosage
cDNA	1 μL
SYBR Green Mix	5 μL
Upstream primers	0.2 μL
Downstream primers	0.2 μL
RNase free H_2_O	3.6 μL

## Data Availability

Not applicable.

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
