# Peer review of "Unraveling the Nrf2-ARE Signaling Pathway in the DF-1 Chicken Fibroblast Cell Line: Insights into T-2 Toxin-Induced Oxidative Stress Regulation"

_toxins, 2023, doi:10.3390/toxins15110627_

Round 1
Reviewer 1 Report
Comments and Suggestions for Authors
The manuscript is interesting and well written but I suggest minor revision of the manuscript based on the following comments:
Comments:
1. In the Abstract and introduction first line, please change “T-2 toxin (T2)” to “T-2 mycotoxin (T2).
2. Please clarify in Figure 2 the idea of 24 h and 48 h (G and H) and (I and J) on the figure itself beside the figure legend. I think it will be more obvious for the reader.
3. In Figure 3, please clarify the unit of malondialdehyde (MDA) on the Figure itself
4. Line 411, please change IC50 into IC50 and where repeated through the manuscript.
5. Please add references 2023, there is only one reference 2023.
6. Line 493, the experiments were repeated three times and the results are reported as the mean ± standard error. I think the standard deviation (SD) is more logic.
Comments on the Quality of English Language
Minor editing
Reviewer 2 Report
Comments and Suggestions for Authors
The present study aimed to investigate the regulatory mechanism of the nuclear factor E2-related factor 2 (Nrf2) against T2 toxin-induced oxidative damage. According to what was reported by the authors, Nrf2 plays a crucial role in the regulation of oxidative stress induced by T2-toxin in the DF-1 chicken fibroblast cell line, by modulating Nrf2 expression through the Nrf2-ARE signaling pathway. In general, this interesting research has important implications in the poultry industry; consequently, the work is worthy of publication in Toxins. However, some minor concerns should be solved before consideration for publication.
Title:
Possible change to: Unraveling the Nrf2-ARE Signaling Pathway in the DF-1 chicken fibroblast cell line: Insights into T-2 Toxin-Induced Oxidative Stress Regulation
Abstract:
L5. Please do not abbreviate T2 toxin, similarly hereinafter.
L5-10. Please delete these lines. You need to insert most observed results in this section of the document.
L16 and L77. Please remove these words: “livestock and”
Key contribution:
L18. Write just one sentence (no more than 5 lines) to describe the most important part of the manuscript.
Introduction:
L33. traction or attraction?
L40. …of reactive oxygen species (ROS)
L68-9. The sentence is rather confusing.
Results:
L86-93. LDH leakage results are missing!
L90. Please avoid the usage of the word “obvious”.
L95. Figure 2, panels I and J. LDH leakage results for the lowest T2 toxin concentration (1 nM) are missing!
L100. The lactate dehydrogenase (LDH) leakage…
Figure 3. Again, results for the lowest T2 toxin concentration (1 nM) are missing!
Figure 3, panels A-E. The quality of the figure is very poor. I strongly recommend the author embellish the figure.
L123. Glutathione peroxidase (GPX) viability?
L255. Figure 9. Please improve the quality of the image (immunoblot).
Discussion:
The discussion of the results is insufficient, and the author’s views need to be supplemented. It is probably due that you start with a general and detailed introduction before a connection to a minor result presentation. I am used to the opposite, which is perhaps more common. One starts with a short results presentation followed by reference to other publications and related discussion.
L374. Please, delete this line.
Conclusions:
More emphasis on findings and its implication may be mentioned in the conclusion section.
Materials and methods:
L394. Dulbecco’s malondialdehyde (DMEM) or Dulbecco’s modified Eagle medium (DMEM)?
L411. The MTT (3-[4,5-dimethylthiazol-2-yl]-2,5 diphenyl tetrazolium bromide) assay…
L412, 422. Cell confluence?
L413. The mycotoxin concentration lacked theoretic support.
L416-18. Not clear what this means. An enzyme marker?
L447-8. Please review the sentence regarding proper syntax.
L458. Please, delete these words “embryonic kidney cells”.
L491-93. Please use the Tukey procedure for means separation. For a better understanding, please add a flow chart of the whole experimental design.
Comments on the Quality of English LanguageThe English style and grammar are satisfactory. Minor editing of English language required.
Reviewer 3 Report
Comments and Suggestions for Authors
The topic is interesting and within the scope of this journal.
In this paper, the authors aim to elucidate the oxidative damage induced by T2 toxin in chicken embryo fibroblast cells (DF-1) with particular attention to the regulatory mechanism of Nrf2-ARE signalling pathway. Overall, the manuscript is well written and suitable for publication after major modifications:
· Page 2, line 65: the word “MAF transcription factor” is bigger, please correct.
· The concentrations of T2 employed were 10, 20, 50 and 100 nM which are very low. Can the authors explain the reason of this choice and introduce in the manuscript?
· Did the authors design the primers for qPCR analysis? And with which database? (If there is any)
· In material and methods section, very few information about RNA extraction and its quality/quantity is included. The extraction procedure should be added, followed by its quantification (total ng/mL) and specifying purity ratios 260/280, 260/230 of each sample.
· Which dye did you use for qPCR reaction? It is necessary to specify. Please check the MIQE guidelines: minimum information for publication of quantitative real-time PCR experiments.
Round 2
Reviewer 3 Report
Comments and Suggestions for Authors
The authors fixed all the recommendation indicated throughout the first revision.